# Selective Survival of Protective Cultures during High-Pressure Processing by Leveraging Freeze-Drying and Encapsulation

**DOI:** 10.3390/foods11162465

**Published:** 2022-08-16

**Authors:** Meghan R. McGillin, Dana L. deRiancho, Timothy A. DeMarsh, Ella D. Hsu, Samuel D. Alcaine

**Affiliations:** Department of Food Science, Cornell University, Ithaca, NY 14850, USA

**Keywords:** HPP, protective cultures, encapsulation, viability, fermentation, lactic acid bacteria

## Abstract

High-Pressure Processing’s (HPP) non-thermal inactivation of cells has been largely incompatible with food products in which the activity of selected cultures is intended (e.g., bio-preservation). This work aims to overcome this limitation using a cocoa butter encapsulation system for freeze-dried cultures that can be integrated with HPP technology with minimal detrimental effects on cell viability or activity capabilities. Using commercially available freeze-dried protective cultures, the desiccated cells survived HPP (600 MPa, 5 °C, 3 min) and subsequently experienced a 0.66-log increase in cell counts during 2 h of incubation. When the same culture was rehydrated prior to HPP, it underwent a >6.07-log decrease. Phosphate-buffered saline or skim milk inoculated with cocoa butter-encapsulated culture up to 24 h before HPP displayed robust cell counts after HPP and subsequent plating (8.37–9.16 CFU/mL). In addition to assessing viability following HPP, the study sought to test the applicability in a product in which post-HPP fermentation is desired While HPP-treated encapsulated cultures initially exhibited significantly delayed fermentative processes compared to the positive controls, by 48 h following inoculation, the HPP samples’ pH values bore no significant difference from those of the positive controls (encapsulated samples: pH 3.83 to 3.92; positive controls: pH 3.81 to 3.85). The HPP encapsulated cultures also maintained high cell counts throughout the fermentation (≥8.95 log CFU/mL).

## 1. Introduction

Increasingly, there is a demand for manufacturing processes that (1) provide foods that are quick and easy to prepare, (2) can ensure food safety, and (3) do not compromise nutritional or sensory qualities [1]. High-pressure processing (HPP) satisfies these criteria for many products, including ready to eat meals, meat, seafood, dairy products, fruits, and vegetables [2]. The exception is when food-grade cultures, such as lactic acid bacteria (LAB), are desired in the final product [3]. At the high-pressure levels used for food preservation (>400 MPa), HPP inactivates pressure-sensitive vegetative cells. This becomes a major limitation for products containing functional microbes, such as probiotic, starter, and protective cultures (Figure 1) [3]. The goal of this work was to develop a novel encapsulation system that renders commercially available freeze-dried LAB resistant to HPP, thus enabling their survival in high moisture food products.

LAB’s dynamic role as a fermentation starter culture extends shelf life, increases microbiological safety, and, in some cases, enhances the sensory and nutritional aspects of the fermented product [4,5,6]. For example, during yogurt production, the fermentation of lactose decreases milk’s pH to below 4.6, resulting in a product that is too acidic for pathogens and spoilage microorganisms to inhabit [4]. This production of organic acids is one example of the antagonistic activity leveraged when LAB are used as protective cultures for bio-preservation [7]. In addition to extended shelf-life, certain strains also confer health benefits to the consumer, demonstrating the virtuous duality of LAB as both probiotic and protective cultures [4]. Many of these desirable attributes are eliminated during treatment due to the indiscriminatory nature of HPPs inactivation mechanism against vegetative cells. Tsevdou et al. [3] found that pressure treatment of yogurt at 400 MPa reduced the counts of probiotic strains below the FDA’s standard for yogurt. This study illustrates the challenges in producing HPP-treated, biologically active food products.

An unexplored opportunity regarding HPP technology is the development of food systems that enable the selective survival of desired functional microbes (i.e., LAB), resulting in food-safe products that display desired microbial attributes (i.e., prolonged shelf-life and probiotic health benefits) (Figure 2). There are isolated cases of species of protective and probiotic cultures that have innate HPP resistance. For instance, it has been demonstrated that when cooked chicken is subjected to HPP, the pressure (400–600 MPa) can act as a selective agent, selecting for *Weisella virdescens*, a non-spoilage LAB with antimicrobial activity [8]. Building off this discovery, Stratakos et al. [9] demonstrated *W. virdescens*’ potential as a protective culture appropriate for HPP-treated foods when they reported *W. virdescens’s* bacteriostatic effects against *Listeria monocytogenes* in HPP-treated (400 MPa) low-acid RTE salads [9]. However, *W. virdescens* failed to reproduce the same protective effect against *Clostridium botulinum* with HPP-treated chicken [10]. These challenges in efficacy highlight the need for greater tools to enable HPP-resistant functional microbes for use in foods.

Pathogenic and spoilage spore-forming organisms are a concern for HPP products since they can survive pressures above 1000 MPa (Figure 1) [11]. *Clostridium* spores pose a serious concern for HPP-treated low-acid foods (pH > 4.6) [11]. To survive inhospitable conditions, spore-forming bacteria adapt their cellular structures by reducing their internal moisture content and coating themselves with additional layers of cell wall. It is believed that their decreased water activity (a_w_) contributes to their ability to withstand extreme conditions, such as high pressure [12]. 

Like sporulation, freeze-drying is a physical process that reduces the a_w_ of cells and may improve a cell’s tolerance to stressful conditions. Kurtmann et al. [13] demonstrated that freeze-dried bacteria were able to survive pressure treatment ranging from 200 to 800 MPa, underlining the protective effect of freeze-drying against high pressures. Though promising, the potential use of such freeze-dried cultures in high-moisture food products, such as hummus or dips, risks the cells rehydrating before the HPP step, thus rendering them susceptible again to high pressure. An encapsulation system is therefore essential for delaying the migration of moisture from the food into the desiccated cells. Although there are numerous studies involving the freeze-drying of cultures after they have been encapsulated, there has been very little research regarding the subsequent encapsulation of pre-existing freeze-dried cultures [14,15,16]. 

Presented here is a novel cocoa butter encapsulation system for freeze-dried cultures that can be used with HPP without detrimental effects on cell viability or acidification capabilities. Overall, the objectives of this project were (1) to demonstrate the protective effect against HPP provided by the freeze-drying of pressure-sensitive protective cultures and (2) to develop an encapsulation system that maintained the desiccated state of freeze-dried cultures when they were used to inoculate a high-moisture food matrix (Figure 2).

To achieve this second aim, commercially available freeze-dried protective cultures (DuPont-Danisco, DuPont-Danisco, Niebüll, Germany) were encapsulated in cocoa butter by homogenization prior to HPP, resulting in cocoa butter-encapsulated freeze-dried cultures. A response surface model (RSM) was generated to optimize the homogenization factors of temperature, speed, and duration. To assess the effectiveness of the cocoa butter barrier in maintaining the desiccated state of the cells, encapsulated cultures were used to inoculate a liquid medium for up to 24 h before HPP. The efficacy of the encapsulation system was determined by post-treatment counts and 48 h fermentation pH curves.

## 2. Materials and Methods

### 2.1. Cultures

The protective culture used in this study was HOLDBAC YM-C Plus LYO 500 DCU (material no. 1285156, DuPont-Danisco, Niebüll, Germany). HOLDBAC YM-C Plus LYO is a freeze-dried multiple-species culture composed of Lacticaseibacillus paracasei and Propionibacterium freudenreichii subsp. shermanii.

### 2.2. HPP Treatment

The experimental designs (Figure 3, Figure 4, Figure 5 and Figure 6) laid out in the following sections reference several instances of HPP treatment. The cultures were subjected to 600 MPa HPP treatment at 5 °C for 3 min using a Hiperbaric 55 (Hiperbaric, Doral, FL, USA) commercial processing unit at the Cornell Food Venture Center Pilot Plant (Cornell AgriTech, Geneva, NY, USA). All HPP-treated cultures were stored at 4 °C following HPP, and all subsequent analyses were carried out within 1 day of HPP.

### 2.3. Assessing the Protective Effect of Freeze-Drying against HPP

To evaluate whether there was a protective effect of freeze-drying against high pressure, HPP-treated freeze-dried (FD.3) and HPP-treated reconstituted cultures’ (RC.3) cell counts were compared to those of freeze-dried cultures that did not undergo HPP (Pos.3) (Figure 3).

#### 2.3.1. Sample Preparation

Two grams of commercial freeze-dried pellets were transferred to a sterile stomacher bag (Whirl-Pak, Madison, WI, USA) and pulverized using a Seward Stomacher Model 400 Circulator Lab Blender (VWR International, Solon, OH, USA) for 1 min at 260 rpm after which the sample was subjected to HPP (FD.3). To compare this treatment to reconstituted cultures at the time of HPP (RC.3), 2 g of freeze-dried pellets were used to inoculate 1× PBS solution (198 mL) for a final ratio of 1:99 (*w*/*v*), and then stomached for 1 min at 260 rpm, prior to HPP. A freeze-dried sample that had not undergone HPP served as a positive control (Pos.3). Uninoculated 1× PBS was included as a negative control (PBS.3). Following HPP, the freeze-dried samples (Pos.3 and FD.3) were used to inoculate in 1× PBS at a ratio of 1:99 (*w*/*v*) and stomached for 1 min at 260 rpm. The four samples (Pos.3, FD.3, RC.3, PBS.3) were transferred to an Isotemp hot plate stirrer (Fisher Scientific, Waltham, MA, USA), where they were suspended using a magnetic stir bar at 810 rpm for 2 h, with heating at 40 °C. Cell counts were determined using the spread plate technique described below under “Enumeration” (Section 2.8).

#### 2.3.2. Statistical Analysis

The experiment was carried out in biological triplicates for statistical purposes. A Student’s *t*-test was performed to identify significant differences between FD.3 and Pos.3. The cell counts for RC.3 were not observable at the limit of detection (3.4 log CFU/mL). This violated the statistical method’s assumption of variance, and therefore this treatment was excluded from the analysis. Statistical analyses were performed using R software (version 4.1.2).

### 2.4. Optimization of Cocoa Butter Encapsulation by Homogenization Using Response Surface Methodology

The freeze-dried cultures are at risk of rehydrating before HPP treatment if applied to a high-moisture food product. To circumvent this, a cocoa butter encapsulation system was developed to delay the migration of moisture from the food into the desiccated cells. To ensure that the encapsulation process would not negatively impact the freeze-dried culture’s viability, an RSM was generated. This RSM provided the experimental design and statistical model for optimizing the encapsulation system’s parameters. Specifically, it investigated the individual and interacting effects of homogenization speed (3000 rpm, 4000 rpm, 5000 rpm), homogenization duration (2 min, 3 min, 4 min), and cocoa butter temperature (30 °C, 35 °C, 40 °C), using a 3-factor second-order Box-Behnken design (BBD) with post-encapsulation cell counts as the response variable. The experimental matrix derived from the BBD outlined fourteen trials varying by levels of homogenization speed, duration, and cocoa butter temperature (Table 1). The design’s two center points were both defined by the mid-level values of the three variables (4000 rpm, 3 min, 35 °C) and served as the only two technical replicates assigned by the matrix. Samples were then prepared in batches based on the 14 trials’ sets of conditions, as generated by the BBD (Table 1).

#### 2.4.1. Sample Preparation

Figure 4 outlines the encapsulation process. For each trial, 2 g of freeze-dried pellets were transferred to a sterile stomacher bag and pulverized using the stomacher for 1 min at 260 rpm. In a similar manner, 18 g aliquots of Greener Life Essentials cocoa butter (Essential Depot, Sebring, FL, USA) were transferred to a 50 mL beaker and heated to 100 °C. The molten cocoa butter was transferred to a water bath and cooled to the temperature designated for that trial by the BBD (30 °C, 35 °C, or 40 °C). The beaker was transferred to a stir plate set to the assigned temperature, where the pulverized culture was added to the cocoa butter at a ratio of 1:9 (*w*/*v*). The culture-cocoa butter solution was thoroughly mixed at 810 rpm for 40 min with heating at the assigned temperature (30 °C, 35 °C, or 40 °C). Each sample of the inoculated cocoa butter was then fully homogenized with an UltraTurrax T 25 (Ika, Wilmington, NC, USA) at the appropriate speed (3000 rpm, 4000 rpm, or 5000 rpm) for the appropriate duration (2 min, 3 min, or 4 min), as assigned by the BBD for the respective trial. The products resulting from this process will subsequently be referred to as “encapsulated cultures” (EN).

#### 2.4.2. Release, Suspension, and Data Collection for Encapsulated Cultures (EN)

The EN samples were used to inoculate 1× PBS at a ratio of 1:9 (*w*/*v*) and suspended in the liquid buffer with stirring at 850 rpm for 2 h with heating at 40 °C. This allowed the cells to release from the cocoa butter matrix and to be reconstituted into their active form. For each trial, the cell counts were determined using the spread plate technique described below under “Enumeration” (Section 2.8). 

#### 2.4.3. Statistical Analysis

The independent variables and their levels used in the response surface design are listed in Table 1. The effects of the independent variables on the log-transformed cell counts were fitted using the RSM function following the second-order model (Equation (1)). The polynomial equation was applied to the RSM to predict the optimal parameters for post-encapsulation viability, as visualized using response plots. The statistical significance of the variables and their interactions was evaluated using a Student’s T-distribution test. Modeling was performed using the RSM (2.10.3) package in R (version 4.1.2).
Y = 11.37 + 0.028X1 + 0.020X_2_ − 0.255X_3_ + 0.027X_1_X_2_ + 0.005X_1_X_3_ +0.065X_2_X_2_ − 0.130X_1_^2^ + 0.023X_2_^2^ − 0.314X_3_^2^(1)

Equation (1). X_1_: homogenization speed; X_2_: homogenization duration; X_3_: cocoa butter temperature; X_1_X_2_:interacting effects of speed and duration; X_1_X_3_:interacting effects of speed and temperature; X_2_X_3_: interacting effects of duration and temperature; X_1_^2^: quadratic effect of speed; X_2_^2^: quadratic effect of duration; X_3_^2^: quadratic effect of temperature.

### 2.5. Encapsulation of Cultures in Preparation for HPP

The encapsulation methodology for the following experiments (Section 2.6 and Section 2.7) followed the procedure detailed above (Section 2.4.1), using the RSM’s predicted optimal conditions. Briefly, cocoa butter was heated to 100 °C, then cooled to 33 °C before adding the freeze-dried culture at a ratio of 1:9 (*w*/*v*). Following stirring using a heated stir plate, the molten cocoa butter inoculum was homogenized (Ultra-Turrax T 25) at 4000 rpm for 3 min. The homogenate was transferred to a sterile stomacher bag and stored overnight at −20 °C, enabling its solidification. The cocoa butter inoculum was then granulated using the shred blade of a model #70740 food processor (Hamilton Beach, Glen Allen, VA, USA) until the granules appeared uniform in size. The granulated EN was then transferred to a new sterile bag and stored at −20 °C.

### 2.6. Application in a High-Moisture Food Matrix

The cocoa butter encapsulated culture’s applicability in high-moisture foods was examined in the following section, as determined by post-treatment viability (Figure 5). In total, eight experimental samples, varying by the following were investigated:the state of encapsulationliquid inoculum mediapre-treatment storage timeHPP treatment.

#### 2.6.1. Sample Preparation

For the HPP-treated samples, the aim was to model potential pre-HPP holding times (24 h vs. 4 h), to determine their effect on post-HPP cell counts, using either a simple buffer solution (1× PBS) or a more complex food matrix (skim milk) as the high-moisture inoculation medium (Figure 5). At 24 h or 4 h before HPP treatment, 0.2 g of EN was transferred to a sterile stomacher bag and used to inoculate 199.8 mL of 1× PBS for a ratio of 1:999 (*w*/*v*). The contents of each bag were then stomached for 1 min at 260 rpm. These two samples will be referred to as EN.6.24.p and EN.6.4.p, respectively. At the same pre-treatment timepoints, two additional EN samples were used to inoculate skim milk (Cornell Dairy, Ithaca, NY, USA) in a likewise manner (EN.6.24.s and EN.6.4.s, respectively). These four samples will be collectively referred to as EN-HPP.6. 

For comparative purposes, unencapsulated freeze-dried cultures were reconstituted prior to HPP to confirm that the encapsulation system is responsible for the presence of viable cultures ensuing HPP. At the selected pre-treatment time points (24 h and 4 h prior to HPP), 1× PBS was inoculated with freeze-dried cultures (1:999 *w*/*v*) and then stomached at 260 rpm for 1 min. These non-encapsulated reconstituted cultures will subsequently be referred to collectively as RC-HPP.6 and individually as RC.6.24 and RC.6.4, respectively. The inoculated samples were stored at 4 °C until HPP. For positive controls, non-HPP-treated encapsulated (EN.6+) and freeze-dried (Pos.6) cultures were used to inoculate 1× PBS at a 1:999 (*w*/*v*) ratio and were mixed thoroughly for 1 min at 260 rpm. Uninoculated skim milk (skim.6) and uninoculated 1× PBS (PBS.6) were included as negative controls. 

#### 2.6.2. Release, Suspension, and Data Collection

Following the HPP step, both the HPP samples and non-HPP controls were transferred to 500 mL glass bottles and placed in a 40 °C water bath for 30 min. This step was included to release any viable cells from the cocoa butter matrix of the EN treatments. Subsequently, each bottle was transferred to a stir plate, where any viable cells were reconstituted by stirring at 850 rpm and heating at 40 °C for 40 min. The samples’ cell counts were then determined using the spread plate technique described below under “Enumeration” (Section 2.8). 

#### 2.6.3. Statistical Analysis

For statistical purposes, all processes outlined in this subsection were carried out in technical triplicates. Log-transformed counts were analyzed using a mixed linear model in R (version 4.1.2), with treatment designated as a fixed effect and repeated trials as a random effect. Pairwise comparisons of the random effects were generated using Kenward-Roger approximated degrees of freedom, using the emmeans (1.7.2) package. The cell counts for the RC-HPP.6 samples (RC.6.24 and RC.6.4) were not observable at the limit of detection (4.4 log CFU/mL). This consistent lack of observable growth violated the model’s assumption of variance, and the RC-HPP.6 samples, therefore, were excluded from the analysis.

### 2.7. Acidification Activity

To determine the impact of HPP on the protective culture’s acidification capacity, pH was measured over a 48 h fermentation, with cell counts determined at 0, 10, 24, and 48 h (Figure 6).

#### 2.7.1. Modifications to Experiment Design Described in Section 2.6

Preliminary trials had utilized pasteurized skim milk as the inoculation medium for these fermentations; however, by 10 h into the fermentation, the skim milk-inoculated samples repeatedly showed evidence of contamination by *Bacillus* spore formers generally known to be native to this substrate (Appendix A). The contaminants were identified as *Bacillus* spp. via polymerase chain reaction amplification of their 16S ribosomal RNA genes, followed by Sanger sequencing (provided by Cornell Biotechnology Resource Center, Ithaca, NY, USA); the resulting sequences were then compared to those in the National Center for Biotechnology Information’s Nucleotide Collection database using the Basic Local Alignment Search Tool. 

Due to the complication of this contamination, skim milk was replaced with sterile Difco lactobacilli de Man, Rogosa and Sharpe (MRS) broth (BD, Franklin Lakes, NJ, USA) as the substrate in the experiment described in this section (Figure 6). Additionally, compared to non-encapsulated culture, the encapsulated culture has already experienced a 1:9 dilution (in cocoa butter); to compensate for this fact, encapsulated cultures were used to inoculate MRS broth at a ratio of 1:99 (*w*/*v*), while non-encapsulated cultures were used to inoculate MRS broth at a ratio of 1:999 (*w*/*v*), to obtain an approximate microbial load of 10 log CFU/mL for both types at the time of immersion.

#### 2.7.2. Sample Preparation

At 24 h or 4 h prior to HPP treatment, 2 g of EN was used to inoculate 198 mL of sterile MRS broth and mixed thoroughly by stomaching for 1 min at 260 rpm. These two EN-HPP.7 samples will hereafter be referred to as EN.7.24 and EN.7.4, respectively. To confirm that the encapsulation system is responsible for the presence of post-HPP viable cultures, MRS broth (199.8 mL) was inoculated with unencapsulated freeze-dried cultures (0.2 g) 24 h and 4 h prior to treatment (RC.7.24 and RC.7.4, respectively). As such, these unencapsulated samples were reconstituted at the time of treatment and collectively referred to as RC-HPP.7. Both the EN-HPP.7 and RC-HPP.7 samples were subjected to HPP following their appropriate inoculation period (24 h and 4 h). To confirm that the protective culture species were absent from the substrate prior to its inoculation, sterile MRS broth was used as a negative control (MRS.7). For positive controls, non-HPP-treated encapsulated cultures (EN.7+) and freeze-dried cultures (Pos.7) were used to inoculate MRS broth at a 1:999 (*w*/*v*) ratio and were mixed thoroughly via stomaching for 1 min at 260 rpm.

#### 2.7.3. Release, Fermentation, and Data Collection

After HPP treatment, the seven samples were transferred into 500 mL bottles. The cultures were released and reconstituted as described above (Section 2.6.2). Next, each bottle was fitted with a sterile lid containing a silicone septum (Corning, Glendale, AZ, USA) through which a sterile InLab Smart Pro-ISM probe (Mettler Toledo; Columbus, OH, USA) was inserted (Figure 7). The probes were attached to an iCinac Fermentation Monitor (AMS Alliance; Rome, Italy), which was programmed to take pH readings every two hours. A sterile 5” hypodermic needle (Air-Tite, North Adams, MA, USA) was also inserted into each bottle through its septum and was attached to a single-use sterile 1 mL Luer-Lok syringe (BD Biosciences; San Jose, CA, USA). These needles remained in the bottles’ septa throughout the fermentation’s duration, during which they served as a means of taking aliquots aseptically for analysis. For each bottle, an airlock was established using a sterile 1000 μL pipette tip and sterile PVC tubing (VWR, Radnor, PA, USA) leading to an output bottle filled with water to allow for any gas to escape. The sample bottles were then transferred to a 40 °C water bath, where they were held for 48 h. To determine cell counts throughout the fermentation, the syringes were used to aspirate aliquots aseptically at 0, 10, 24, and 48 h. To obtain cell counts, these aliquots were then enumerated as described in Section 2.8.

#### 2.7.4. Statistical Analysis

For statistical purposes, all processes outlined above were carried out in technical triplicates. Applying Kenward-Roger approximated degrees of freedom, pH and log-transformed cell counts were each analyzed using a mixed linear model, with treatment, timepoint, and the interaction between treatment and timepoint designated as fixed effects and repeated trials as a random effect. The cell counts for the RC-HPP.7 samples (RC.7.24, RC.7.4) and negative control (MRS.7) were not observable at the limit of detection (4.4 log CFU/mL). This consistent lack of observable growth violated the statistical model’s assumption of variance, and the RC-HPP.7 samples and MRS.7 control were excluded from the analysis. Statistical analyses were performed using R (version 4.1.2). 

### 2.8. Enumeration

At each point indicated in the above procedures, a representative aliquot of each sample was subjected to serial dilution, and 100 μL of each of a set of appropriate dilutions was plated onto MRS agar; plating was carried out in duplicate, and plates were incubated at 30 °C for 48 h. Duplicate plates’ CFU were counted using a Chemopharm Color QCount model 530 (Advanced Instruments, Inc., Norwood, MA, USA). The following settings were utilized in Qcount’s software: Source: Spread; Minimum size: 0.10 mm; Maximum size: 20.00 mm; Shutter speed: 1/125; Lighting: bottom; True; Grid: circular; Sample volume: 0.1 mL; Area multiplier: 5.00%; Low count: 20 colonies; High count: 300 colonies.

## 3. Results

### 3.1. Freeze-Dried Cultures Survived HPP

If the desiccated state of the freeze-dried culture does confer a protective effect against HPP, that would be reflected in high cell counts following treatment. The HPP-treated reconstituted cultures (RC.3) showed no growth at the minimum limit of detection (3.4 log CFU/mL) following treatment. This represented a >6.07-log decrease in cell counts across the three trials for the RC.3 samples. There was a significant (*p* < 0.001) 1.23-log reduction in FD.3 (10.13 log CFU/mL) compared to Pos.3 (11.36 log CFU/mL) (Figure 8). However, FD.3’s final cell count still represents a 0.66-log increase above the cell concentration of this sample upon the event of its inoculation prior to HPP (9.47 log CFU/mL), indicating the viability of this culture following HPP. Although the freeze-dried state of the protective culture did not shield it entirely from the deleterious effects of high pressure, it did appear to provide enough of a protective effect to enable the culture to survive in concentrations sufficient for subsequent inoculations of food matrices.

### 3.2. Encapsulation in Molten Cocoa Butter by Homogenization Did Not Impact the Protective Culture’s Viability

Maintaining the desiccated state of the freeze-dried culture prior to and during processing presents a challenge in high-moisture HPP foods. Therefore, an encapsulation system that prevents the migration of moisture from the food matrix to the desiccated cells until after processing is needed to circumvent this issue. To fulfill this requirement, a cocoa butter-based encapsulation system was designed and optimized using a RSM. A 3-factor BBD examined the effects of homogenization speed (X_1_), homogenization duration (X_2_), and cocoa butter temperature at the time of encapsulation by homogenization (X_3_) (Table 1).

#### 3.2.1. Encapsulation Yields

Overall, cocoa butter encapsulation yielded cell counts comparable to those of the non-HPP freeze-dried control. The generated design’s center points (4000 rpm, 3 min, 35 °C) were associated with the highest observed cell counts of 11.40 and 11.34 log CFU/mL, respectively. In contrast, the lowest viability (10.57 log CFU/mL) was associated with the homogenization of culture with 40 °C molten cocoa butter at 3000 rpm for 3 min. Low viability was observed for all the factor level combinations assigned a cocoa butter temperature of 40 °C. Collectively, there was less than a log difference between the lowest and highest observed average counts (10.57 to 11.40 log CFU/mL) of the EN samples.

#### 3.2.2. Response Surface Model and Statistical Significance of Temperature

The non-linear effects of homogenization speed and cocoa butter temperature on post-encapsulation counts are depicted in Figure 9. When homogenization duration is held constant at 3 min, the temperature’s effect is skewed such that lower temperatures result in higher counts, with the peak occurring at 33 °C. Comparatively, speed’s effect is more centralized, with the maximum response occurring at 4000 rpm. Statistical analysis of the model supported the inverse relationship between high temperatures and lower cell counts. The high negative estimated value for temperature’s effect and its high level of significance (*p* < 0.001) signaled the importance of the molten cocoa butter temperature during the encapsulation process, emphasizing the deleterious effects of higher temperatures on cell viability. The quadratic effect for this variable (X_3_^2^) was smaller but significant when assessed at a higher alpha cut-off (*p* < 0.05), indicating temperature’s negative quadratic effect on EN’s counts. Collectively, the significant linear and quadratic effects resulted in an overall curvilinear effect (Figure 9). Consistent with the response plots and observed cell counts, the stationary points of the model predicted that encapsulation in molten cocoa butter at 33 °C and homogenization at 4105 rpm for 3 min would yield the highest cell viability. 

### 3.3. Encapsulation Prevents Unintended Rehydration of Cultures up to 24 h before HPP Treatment

Inoculating a high-moisture medium (PBS or skim milk) with the encapsulated cultures prior to HPP yielded high cell counts following HPP treatment (Figure 10). EN.6.24.p and EN.6.4.p averaged 8.37 ± 0.26 and 8.98 ± 0.13 log CFU/mL, respectively (Table 2). Similarly, EN.6.24.s and EN.6.4.s averaged 8.6 ± 0.36 and 9.16 ± 0.25 log CFU/mL, respectively (Table 2). Differences in inoculation medium or pre-treatment inoculation period yielded no significant differences in cell counts in the EN-HPP.6 samples (*p* < 0.05). RC.6.24 and RC.6.4 showed no growth at the minimum limit of detection (4.4 log CFU/mL) following treatment across all three trials and were excluded from statistical analysis. Table 2 lists the mean log counts for the encapsulated and non-encapsulated cultures and their standard errors. Pairwise comparisons of cell counts revealed that the EN-HPP.6 samples (EN.6.24.p, EN.6.4.p, EN.6.24.s, and EN.6.4.s) underwent significant (*p* < 0.05) 1-to-3 log reductions in viability when compared to Pos.6 (11.16 ± 0.38 log CFU/mL) and EN.6+ (10.86 ± 0.14 log CFU/mL). 

### 3.4. HPP-Treated Encapsulated Cultures Maintain Acidification Activity in Liquid Food Matrix throughout 48-h Fermentation

The EN-HPP.7 samples (EN.7.24 and EN.7.4) demonstrated functional acidification activity following HPP treatment compared to the non-HPP positive controls (Pos.7 and EN.7+). The samples subjected to a 24 h pre-HPP inoculation period (EN.7.24) appeared to have delayed activity when compared to samples with the shorter inoculation period (EN.7.4) or non-HPP treated samples (Pos.7 and EN.7+) (Figure 11A). At the beginning of the fermentation, there were no significant differences between the pH values of the EN-HPP.7 samples and those of the positive controls (Pos.7 and EN.7+) (Table 3). By 10 H, EN.7.24 displayed a significantly higher pH compared to the pH values for EN.7.4 and the positive controls (EN.7+ and Pos.7). By 24 h, the pH for EN.7.24 decreased by one unit, bearing no significant differences to the pH values of EN.7.4 and the positive controls (Table 3). The pH values of the EN-HPP.7 samples and the positive controls all fell below the pH 4.6 threshold by the end of the 48-h fermentation (Figure 11A). Figure 11B shows that EN-HPP.7 samples maintained high cell counts throughout the 48-h fermentation, despite starting with lower counts than the positive controls. At the beginning of the fermentation (T = 10 h), there were significant differences between the cell counts of EN.7.24 and EN.7.4 when compared to those of the controls (EN.7+ and Pos.7), but those differences lost their significance after the 10 h mark (Table 3). RC.7.24 maintained a stable pH throughout the fermentation. However, the pH of both RC.7.4 and MRS.7 decreased by approximately 0.5 units throughout the 48-h fermentation. Their final pH readings remained above the 4.6 threshold (pH 5.95 and pH 5.84, respectively) and were significantly higher than the pH readings associated with the EN-HPP.7 samples and positive controls at that time point (Figure 11). Regardless, the RC-HPP.7samples (RC.7.24 and RC.7.4) and the MRS.7 control did not show observable growth at the minimum limit of detection (4.4 log CFU/mL) at any point throughout the fermentation (Table 3). 

#### 3.4.1. Fermentation at T = 0 h

Collectively, the initial pH readings amongst samples ranged by 0.77 units at T = 0 h, with MRS.7 starting with the highest pH of 6.39 and Pos.7 with the lowest pH of 5.62 (Table 3). EN.7+ had the second-lowest pH of 5.87. Between the EN-HPP.7 samples, a lower pH was recorded for EN.7.24 (6.18) compared to EN.7.4 (6.3); however, these differences were not statistically significant (*p* < 0.05). Likewise, the pH values for both positive controls (Pos.7 and EN.7+) were not significantly different from either of the EN-HPP.7 samples. The pH of Pos.7 was significantly different from those of both RC.7.4 (6.37) and the MRS.7 control (6.39) (*p* < 0.05). RC.7.24’s initial pH (5.95) was not significantly different from those of any of the HPP-treated samples nor of the controls (Pos.7, EN.7+, MRS.7). 

At T = 0 h, MRS.7 and the RC-HPP.7samples showed no observable growth at the limit of detection (4.4 log CFU/mL) and therefore were excluded from the statistical analysis. Cell counts for the treatments and controls at T = 0 h all fell within a 2.33-log range, with Pos.7 and EN.7+ yielding significantly higher counts (11.28 log CFU/mL and 11.16 log CFU/mL, respectively) than EN.7.24 and EN.7.4 at this time (*p* < 0.05) (Table 3). Between the encapsulated HPP-treated samples, EN.7.4 cell counts were higher (10.07 log CFU/mL) than those of EN.7.24 (8.95 log CFU/mL) but were not significantly different (*p* < 0.05). 

#### 3.4.2. Fermentation at T = 10 h

Across samples, the pH curves began to diverge within 10 h of the fermentation, with the pH range between the most acidic (4.17, Pos.7) and least acidic (6.37, RC.7.4) sample expanding to 2.2 pH units by T = 10 h (Figure 11). The pH reading for EN.7.4 fell below EN.7.24’s pH reading at that time (4.59 vs. 5.33, respectively), despite EN.7.4 starting with a higher initial pH (6.3 vs. 6.18, respectively). At T = 10 h, a trend emerges, with EN-HPP.7 samples and positive controls having significantly different pH measurements overall from those of RC-HPP.7 samples and MRS.7 (Table 3). The exception to this trend was EN.7.24, which had a significantly different pH compared to all the samples besides RC.7.24. Likewise, there were significant differences in pH measurements for MRS.7 and RC.7.4 when compared to the EN-HPP.7 samples and the positive controls.

By T = 10 h, MRS.7 and the RC-HPP.7 samples (RC.7.24 and RC.7.4) showed no observable growth in cell counts at the limit of detection (4.4 log CFU/mL) and were therefore excluded from the statistical analysis. The range between the lowest (9.73 log CFU/mL, EN.7.24) and highest (10.87 log CFU/mL, EN.7+) counts decreased by 1.19 logs, with all the EN-HPP.7 samples and positive controls spanning within 1.14 logs of each other (Figure 11B). This convergence can be attributed to a reduction in counts for Pos.7 and EN.7+ and an increase in counts for the EN-HPP.7 samples occurring over this period. Pos.7 (10.34 log CFU/mL) underwent the most pronounced change over this period, with a 0.94-log reduction in CFU/mL. During this period, the EN.7+ (10.87 log CFU/mL) and EN.7.4 (10.71 log CFU/mL) surpassed Pos.7 (10.34 log CFU/mL) in counts (Table 3); however, the differences in counts across the EN-HPP.7 samples and positive controls were determined to be statistically insignificant (*p* < 0.05).

#### 3.4.3. Fermentation at T = 24 h

Midway through the fermentation, the EN-HPP.7 samples (EN.7.24 and EN.7.4) and positive controls (EN.7+ and Pos.7) had all reached a pH below 4.6 (Figure 11A). No significant differences in pH were identified between the EN-HPP.7 samples and the positive controls at this timepoint (Table 3). EN.7+ had the lowest pH measurement of 3.93, rather than Pos.7, which trailed closely behind with a pH of 3.97. EN.7.24 had a pH of 4.36, and EN.7.4 had a pH of 4.07. By T = 24 h, differences in pH narrowed across the group of samples, including the EN-HPP.7 treatments and the positive controls, with pH values spanning a range of 0.43 units between the highest (EN.7.24) and lowest (EN.7+) readings at this time. When all treatments and controls are considered together, a widening gap in pH values was observed, with the range between the highest pH (6.20, RC.7.4) and lowest pH (3.93, EN.7+) spanning 2.27 units (Figure 11). At T = 24 h, the pH values of the RC-HPP.7 samples and MRS.7 control were significantly different from those of the EN-HPP.7 samples and positive controls (Table 3).

MRS.7 and the RC-HPP.7 samples continued to show no observable growth at the limit of detection (4.4 log CFU/mL) at T = 24 h. The counts for EN.7+ and Pos.7 continued to decline (10.61 log CFU/mL and 9.8 log CFU/mL, respectively), while the counts for the EN-HPP.7 samples peaked at 10.70 log CFU/mL for EN.7.24 and 11.00 log CFU/mL for EN.7.4 (Figure 11B). EN.7.24 displayed the most pronounced change during this period, with a 0.97-log increase over 14 h, whereas the Pos.7 control showed the greatest reduction (-0.54 log CFU/mL) during this time. Overall, there was a 1.2-log difference between the highest (EN.7.4) and lowest (Pos.7) counts, with no significant difference observed across the EN-HPP.7 samples and positive controls (Table 3). 

#### 3.4.4. Fermentation at T = 48 h

By the end of the fermentation, the EN-HPP.7 samples and positive controls converged within a pH range below 4 (Figure 11A). Consistent with T = 24 h, the final pH readings of the HPP-treated encapsulated samples, and the positive controls were not significantly different from each other (Table 3). The pH values of these samples at this timepoint fell within a range of 0.11 units, with the lowest final pH of 3.81 associated with the EN.7+ and the highest associated with EN.7.24 (3.92). EN.7.4 had the second-lowest pH (3.83), followed by Pos.7 (3.85). By the end of the fermentation, the pH values for the RC-HPP.7 samples and MRS.7 remained significantly different compared to those of the EN-HPP.7 samples and positive controls. At this time point, the range between the highest (RC.7.4, 5.95) and lowest (EN.7+, 3.81) pH values spanned 2.14 units.

MRS.7 and the RC-HPP.7 samples continued to show no observable growth at the limit of detection (4.4 log CFU/mL) by the end of the 48-H fermentation. The cell counts across the EN-HPP.7 samples and positive controls concentered between 9–10 log CFU/mL, with all of them falling within 0.59 logs of each other (Figure 11B). EN.7.4 (9.57 log CFU/mL) underwent the most pronounced decline during the final 12 h of fermentation, with a reduction of 1.43 log CFU/mL, followed by EN.7.24 (9.5 log CFU/mL), with a 1.2-log reduction. Pos.7 (9.35 log CFU/mL) and EN.7+ (9.94 log CFU/mL) continued to decline in counts, but less severely than did the HPP-treated samples (−0.45 and −0.67 log CFU/mL, respectively). The difference in cell counts across the EN-HPP.7 samples, and positive controls remained insignificant at the end of the fermentation (Table 3).

## 4. Discussion

This study seeks to create HPP food systems that enable the selective growth of desired functional cultures while also inactivating pathogens and spoilage organisms. Such systems address an unmet need in the food industry, and their creation would expand the market for HPP-treated probiotic and fermented foods. Protective cultures could be incorporated into preservation methods preceding HPP treatment, resulting in a double hurdled HPP and bio-preservation technique. This would allow live cultures to be applied to pre-packaged products, enabling the HPP of pre-sealed food products while maintaining the desired microbial activity, thus reducing the risk of secondary contamination following processing (Figure 2).

The goal of this work was to enable the survival of freeze-dried protective cultures to HPP and thus extend HPP food applications. In fulfilling the first aim of this experiment, we demonstrated that protective cultures could survive HPP when freeze-dried but regained susceptibility to high pressures if reconstituted in a high-moisture medium prior to processing (Figure 8). Compared to non-HPP-treated cultures (Pos.3), there was a significant 1.23-log reduction in cell counts of the FD.3 sample following treatment; nonetheless, the FD.3 samples survived treatment at sufficiently high concentrations (10.13 log CFU/mL) to provide a potentially protective effect. 

An encapsulation system that protects the desiccated cells from rehydrating (by providing a lipid barrier against the migration of moisture from the food matrix) is essential for the application of freeze-dried protective cultures in high-moisture HPP-treated foods. To delay rehydration, freeze-dried cultures were encapsulated in cocoa butter via homogenization (Figure 2). This system was optimized using an RSM that analyzed the effects of homogenization speed and duration and cocoa butter temperature on cell viability. The model predicted that homogenizing the cultures at 4000 rpm for 3 min in 33 °C cocoa butter would produce the highest cell counts. The only significant factor (*p* < 0.05) affecting cell viability was cocoa butter temperature, with lower temperatures yielding higher counts (Figure 9). After implementing the model’s optimal parameters, the encapsulation process was not shown to impact cell viability, with the encapsulated cultures yielding counts similar to those of non-encapsulated cultures (11.4 log CFU/mL). 

To determine the effectiveness of the encapsulation system when applied to high-moisture HPP products, the encapsulated cultures were used to inoculate a liquid medium (skim milk, PBS buffer, or MRS broth) before HPP treatment. As it is commonplace in industry to outsource HPP to third-party providers, factors like transportation and pre-treatment storage periods need to be considered when assessing the effectiveness of the encapsulation system. To gauge the range in which the encapsulation system is effective based on realistic storage times, a high-moisture medium was inoculated with the encapsulated samples at either 24 h or 4 h before treatment. PBS or skim milk was selected as the liquid inoculum to assess the application of this system in a simple buffer compared to a more complex food milieu.

Within the EN-HPP.6 samples (EN.6.24.p, EN.6.4.p, EN.6.24.s, EN.6.4.s), there were no significant differences (*p* < 0.05) in cell counts between inoculation periods (24 h vs. 4 h), nor between media (PBS vs. skim milk) (Table 2). Compared to non-treated positive controls, inoculated in 1× PBS 24 h prior to HPP (EN.6.24.p) yielded the greatest reduction in counts across all EN-HPP.6 samples, with 2.79- and 2.49-log reductions compared to the counts of Pos.6 and EN.6+, respectively (Figure 10). Skim milk inoculated 4 h prior to HPP (EN.6.4.s) yielded the lowest reduction in counts in comparison to the positive controls, with 2.00- and 1.70-log reductions compared to the Pos.6 and EN.6+, respectively. 

Unsurprisingly, encapsulated samples subjected to longer pre-treatment inoculation periods (24 h) yielded lower counts compared to the 4 h pre-treatment inoculation period; however, these differences were not significant (Table 2). Similarly, the inoculation medium appeared to have no significant effect on post-treatment viability (Table 2). 

Overall, our results demonstrate that the encapsulated cultures can be used to inoculate PBS or skim milk for up to 24 h prior to treatment and can still maintain sufficiently high counts post-HPP (>8 log CFU/mL).

Acidification of the food matrix is one of the inhibitory mechanisms provided by the protective cultures used in this study. To ensure that the treated cultures maintained their ability to acidify the food matrix while maintaining competitively high counts, pH and cell counts were monitored over a 48-h fermentation. A pH below 4.6 is deemed too acidic for pathogenic spore formers like *C. botulinum* and served as the threshold for what was considered a successful fermentation for this experiment [17].

Preliminary fermentations using skim milk as the fermentative inoculation medium were disrupted by *Bacillus* spore former contamination by the 10 h time point of the fermentation, prompting the replacement of skim milk with MRS broth in subsequent fermentations (Appendix A). This issue of contamination highlights the variability in protective cultures’ efficacy, which is dependent on strain and species, as well as on the extent of contamination [18]. The protective mechanisms of the cultures used in this study are partially due to their acidification abilities; therefore, it is not surprising that they were ineffective at inactivating *Bacillus* spores. Numerous species within the *Bacillus* genus are lactic acid-producing and thus tolerant to acidic conditions [19]. However, this is not the case for spore formers within the *Clostridium* genus, which are inactivated at a lower pH (4.6), demonstrating the potential application of this system for the specific control of *Clostridium* spore formers [17,20].

In MRS broth, EN-HPP.7 samples maintained their acidification abilities and demonstrated high cell counts throughout the fermentation (Figure 11). Despite the EN-HPP.7 cultures’ prolonged inoculation in MRS broth prior to fermentation, their initial pH values remained above 6, indicating that the EN-HPP.7 samples were not actively fermenting the product during the pre-HPP storage time (Figure 11A). Although the pH values for EN.7.4 and EN.7.24 had significantly differed at T = 10 h, by halfway through the fermentation (T = 24 h), EN.7.24 converged with EN.7.4 and the positive controls; this group bore no significant differences at this time point. By 48 H, the pH readings for the EN-HPP.7 samples and positive controls fell to an acidic pH below 4; the difference between the mean pH values of these two samples was statistically insignificant.

The final pH readings (T = 48 h) for the RC-HPP.7 samples and MRS.7 were significantly different from those of the EN-HPP.7 samples and the positive controls (Table 3). A small reduction in pH was observed after the 10 h time point for both RC.7.24 and MRS.7; however, neither sample’s pH fell below the 4.6 threshold by the end of the fermentation, and both samples’ final pH measurements were significantly higher than those of the EN-HPP.7 samples and positive controls. The MRS plates corresponding to the RC-HPP.7 samples and MRS.7 presented no growth of the protective cultures at the minimum level of detection (4.4 log CFU/mL) across all three trials, though there were some isolated cases of *Bacillus* spore-former growth (too few to count (TFTC), <30 CFU) on the lowest dilution plated for each of these samples. The source of this environmental contaminant was not determined, but it only appeared on the RC-HPP.7 and MRS.7 plates, supporting the hypothesis that the pH activity observed with the EN-HPP.7 samples and positive controls was due to the protective cultures rather than the result of environmental contamination.

For the EN-HPP.7 samples and positive controls (Pos.7 and EN.7+), high cell counts (>8 log CFU/mL) were observed throughout the fermentation (Figure 11B). Despite the initial (T = 10 h) significant differences in cell counts between the positive controls and the EN-HPP.7 samples, the differences in counts were no longer significant at T = 10 h and afterward (Table 3). By 24 h, the counts for the EN-HPP.7 samples peaked; these high counts were like those observed for the positive controls at their highest concentrations, both of which occurred at the beginning of the fermentation. By the end of the fermentation (T = 48 h), counts for the EN-HPP.7 samples and the positive controls all fell within a range of 9–10 log CFU/mL (Table 3). 

For applications with protective cultures, future work is needed to assess the inhibitory effect of this system against Clostridium spores. For applications in fermented and probiotic products, future work is needed to examine the effects of longer-term storage on selected culture viability following HPP treatment.

## 5. Conclusions

Collectively, this work shows that a simple cocoa butter encapsulation system for freeze-dried protective cultures can be used in combination with HPP without detrimental effects on cell viability or acidification capacities. Cocoa butter encapsulation was demonstrated to be a reliable system for preserving the viability and functionality of protective cultures. This was confirmed using both plate enumeration and monitoring of pH change over time. An RSM was utilized to optimize cell viability against several relevant factors during the encapsulation process; the only factor deemed significant was the temperature of the cocoa butter, with lower temperatures yielding higher cell counts. Pre-treatment inoculation period and substrate type were also shown to be nonsignificant factors regarding cell survival. Treatments where protective cultures were used to directly inoculate a substrate without encapsulation were unable to survive HPP. This research may provide an opportunity to overcome the current limitations of HPP in allowing for the production of HPP-treated fermented foods that maintain their probiotic qualities (Figure 2).

## Figures and Tables

**Figure 1 foods-11-02465-f001:**
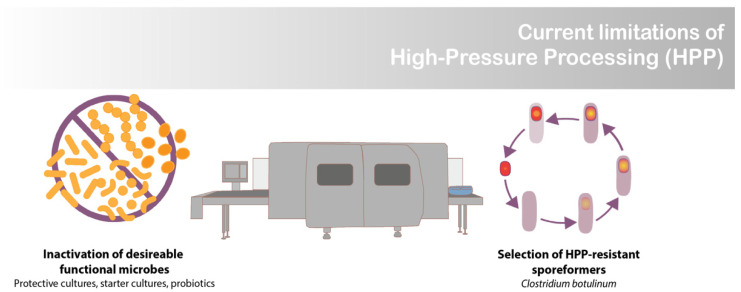
Limitation of high-pressure processing (HPP).

**Figure 2 foods-11-02465-f002:**
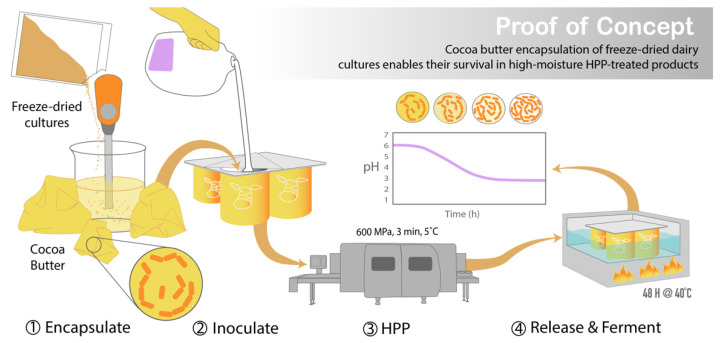
Graphic abstract illustrating the development of this encapsulation system and its potential application in HPP-treated foods with functional cultures. Following cocoa butter encapsulation of the freeze-dried cultures via homogenization, the encapsulated cultures were used to inoculate milk in a packaged container and sealed prior to HPP treatment. After processing, the cultures are released from the cocoa butter through the application of heat (40 °C), prompting the fermentation of the milk by the cultures into yogurt within the sealed packaged container.

**Figure 3 foods-11-02465-f003:**
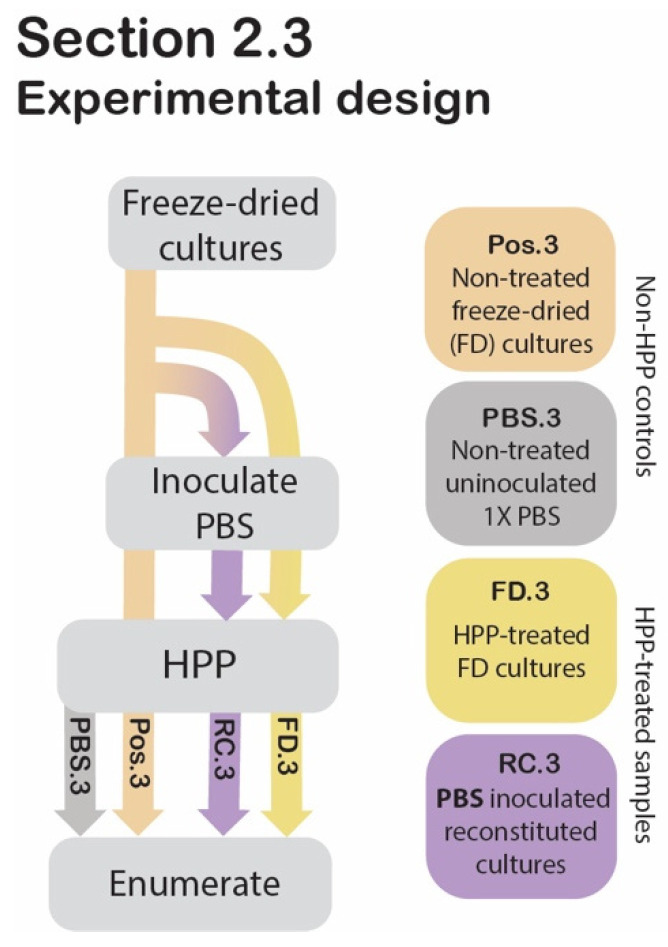
Section 2.3 experimental design. Process flow diagram displaying the step-by-step procedure of preparation and evaluation of post-HPP survival of freeze-dried and reconstituted cultures compared to non-HPP controls. Arrowheads indicate that the proceeding step was applied to the sample, while lines that pass behind the text box indicate that the step was bypassed for the sample.

**Figure 4 foods-11-02465-f004:**
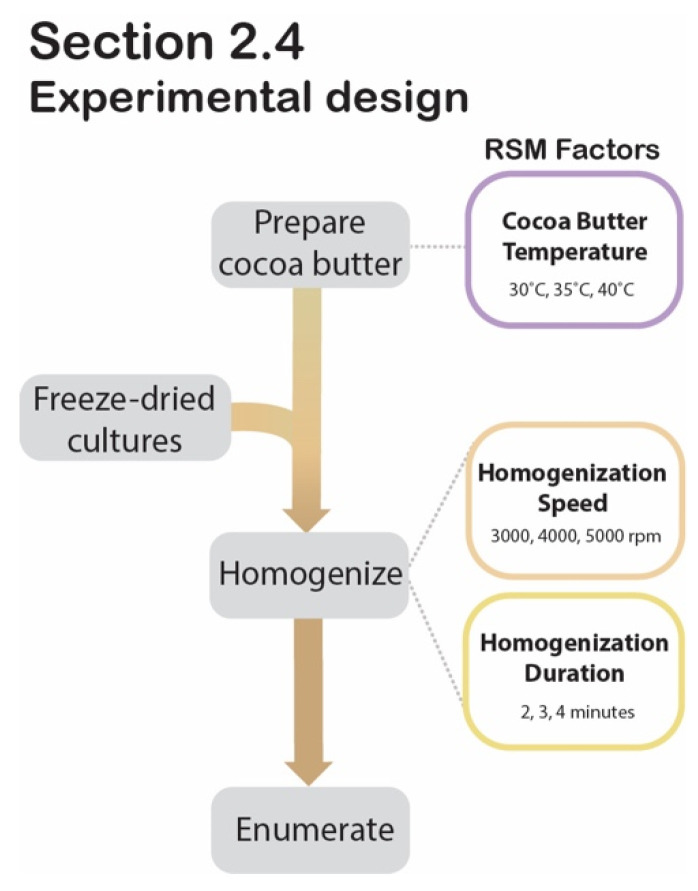
Section 2.4 experimental design. Process flow diagram displaying the step-by-step procedure for RSM experimental design to determine significant factors during cocoa butter encapsulation.

**Figure 5 foods-11-02465-f005:**
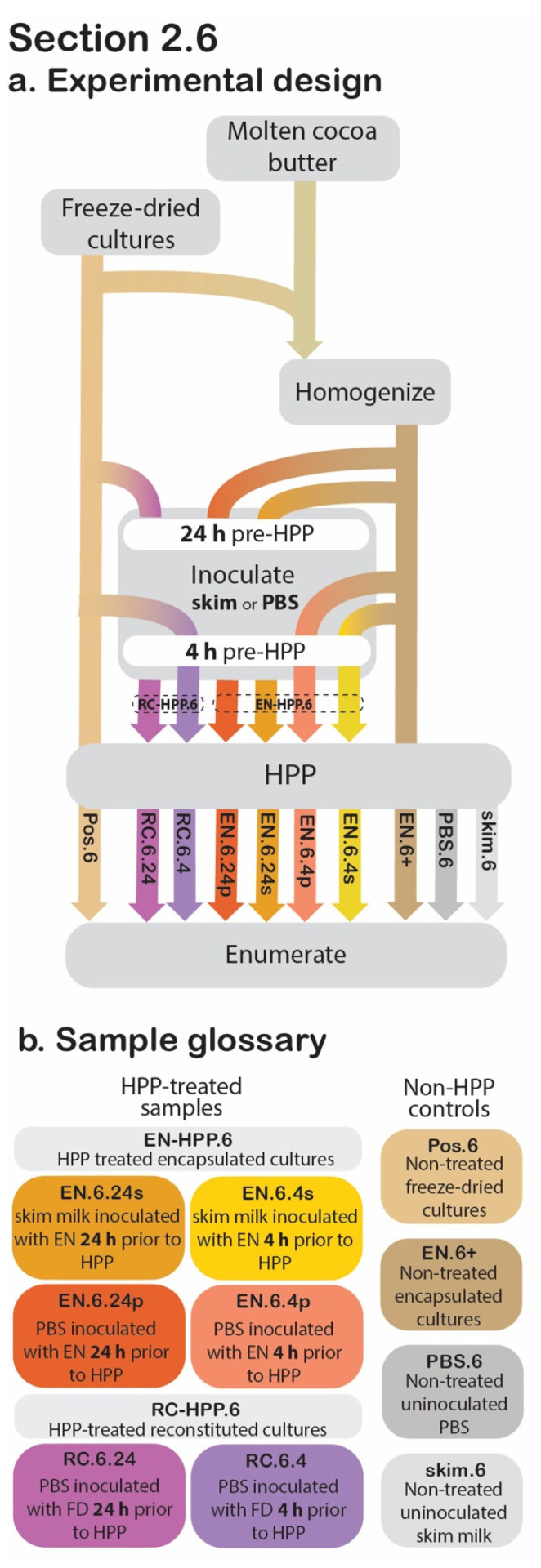
Section 2.6 experimental design. Process flow diagram displaying the step-by-step procedure of preparation and evaluation of post-HPP survival of cocoa butter encapsulated cultures. Arrowheads indicate process step was applied to that sample, whereas continuous lines behind the process step (grey box) indicate that it was not applied to that sample (**a**). The sample glossary defines treatments specific to this section (**b**).

**Figure 6 foods-11-02465-f006:**
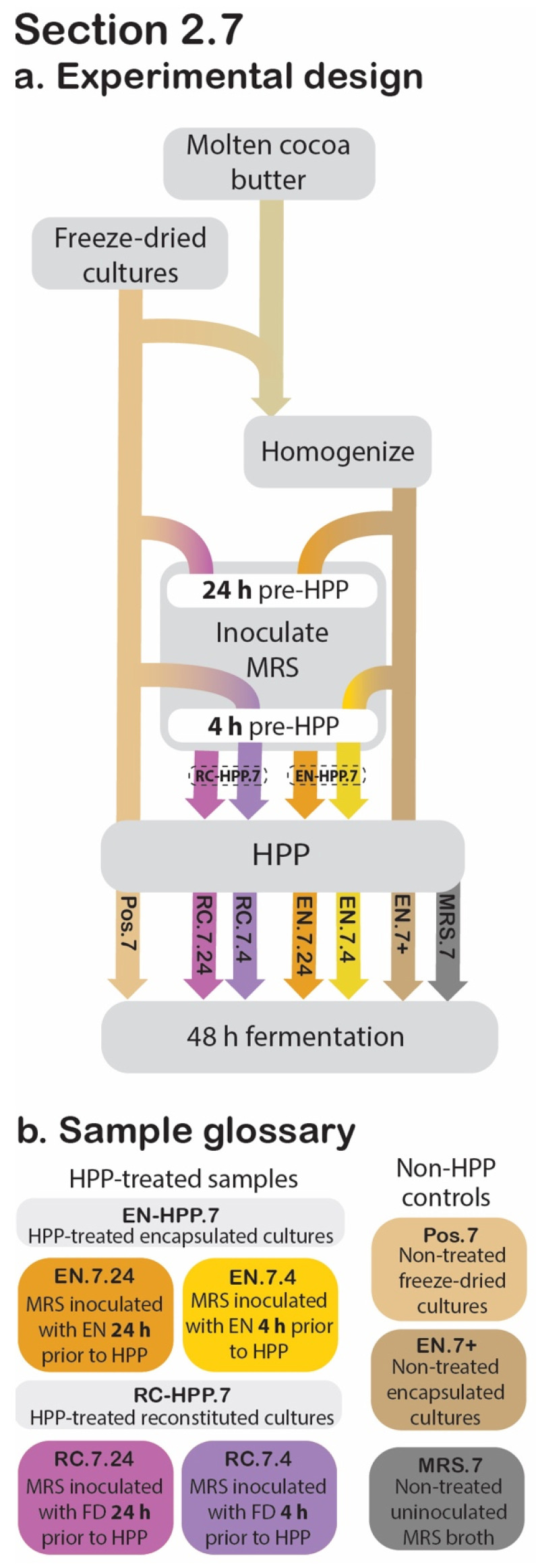
Section 2.7 experimental design. Process flow diagram displaying the step-by-step procedure used to evaluate the retained acidification abilities for the HPP-treated encapsulated cultures (EN) and reconstituted cultures (RC) in relation to the positive and negative controls. Arrowheads indicate process step was applied to that sample, whereas continuous lines behind the process step (grey box) indicate that it was not applied to that sample (**a**). The sample glossary defines treatments specific to this section (**b**).

**Figure 7 foods-11-02465-f007:**
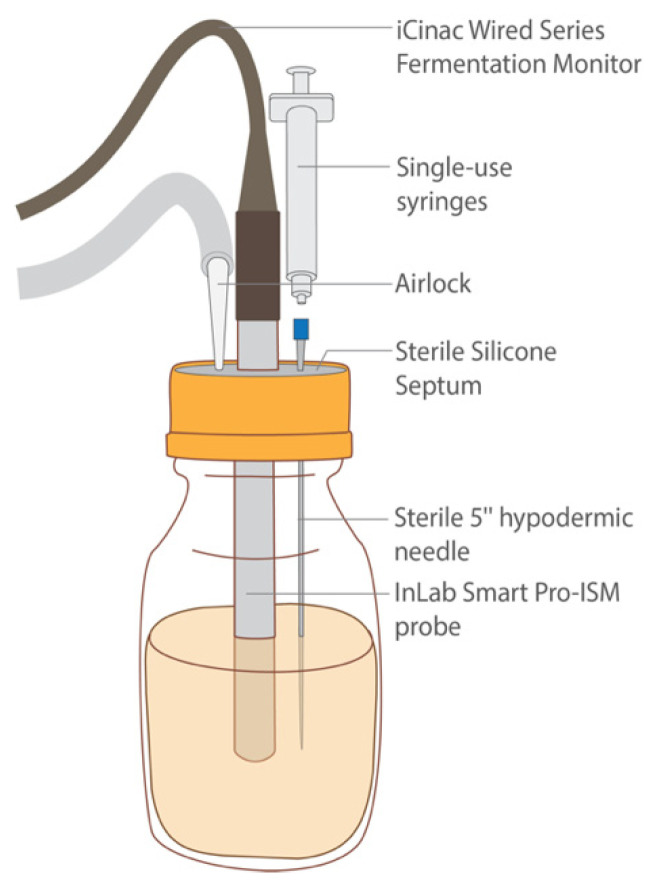
Diagram of fermentation sample depicting the iCinac probe for pH monitoring, the hypodermic needle and syringe for sample extraction, and the airlock system for excess gas production. All these components were secured in the bottle aseptically using a sterile septum.

**Figure 8 foods-11-02465-f008:**
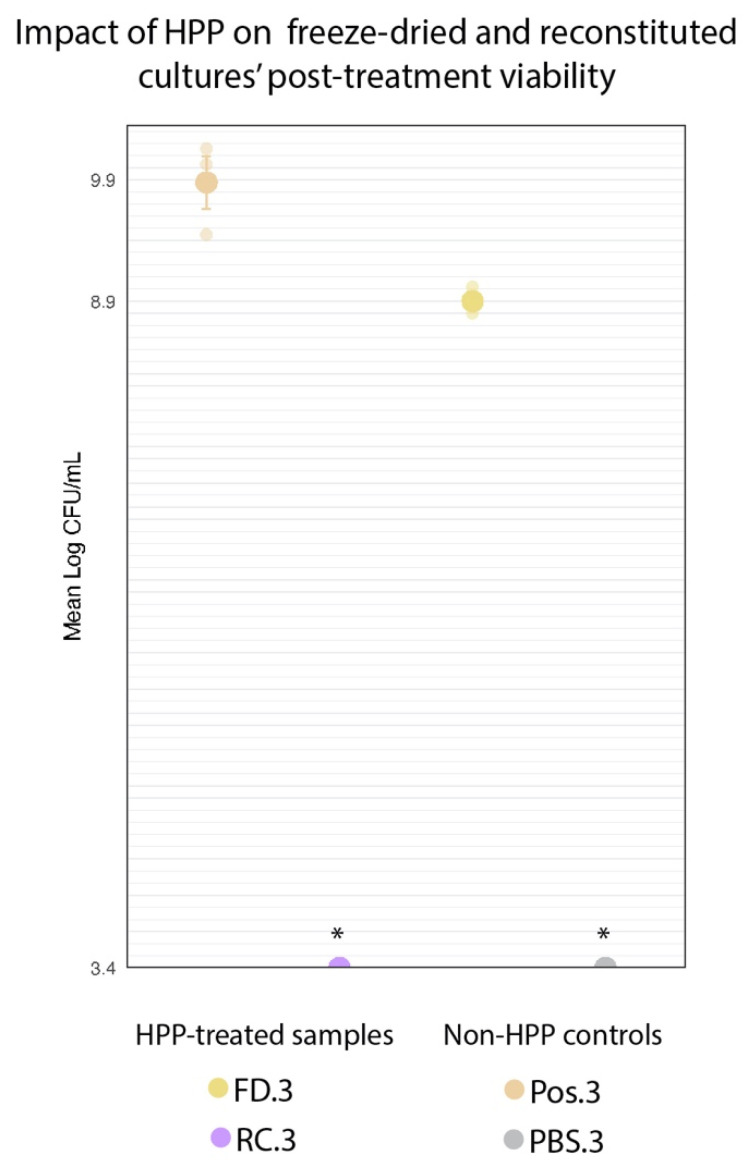
Comparison of log-transformed cell counts of HPP-treated freeze-dried cultures (FD.3) and non-HPP-treated freeze-dried positive control (Pos.3). Error bars represent standard error (SE). * RC.3 and PBS.3 counts were below the level of detection (3.4 log CFU/mL).

**Figure 9 foods-11-02465-f009:**
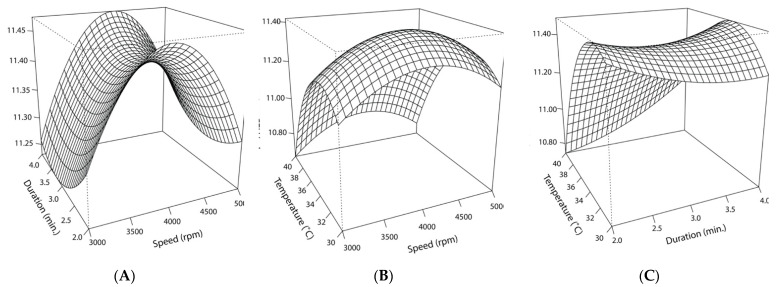
Response surface curve for the interacting effects of speed (rpm), duration (min.), and temperature (°C) on cell counts following encapsulation in cocoa butter via homogenization: (**A**) The predicted interacting effects of homogenization speed (X_1_) and duration (X_2_) while holding the effect of cocoa butter temperature constant (35 °C); (**B**) the predicted interacting effects of homogenization speed (X_1_) and cocoa butter temperature (X_2_) at the time of homogenization while holding the effect of homogenization duration constant (3 min.); (**C**) the predicted interacting effects of homogenization duration (X_2_) and cocoa butter temperature (X_3_) at the time of homogenization while holding the effect of homogenization speed constant (4000 rpm).

**Figure 10 foods-11-02465-f010:**
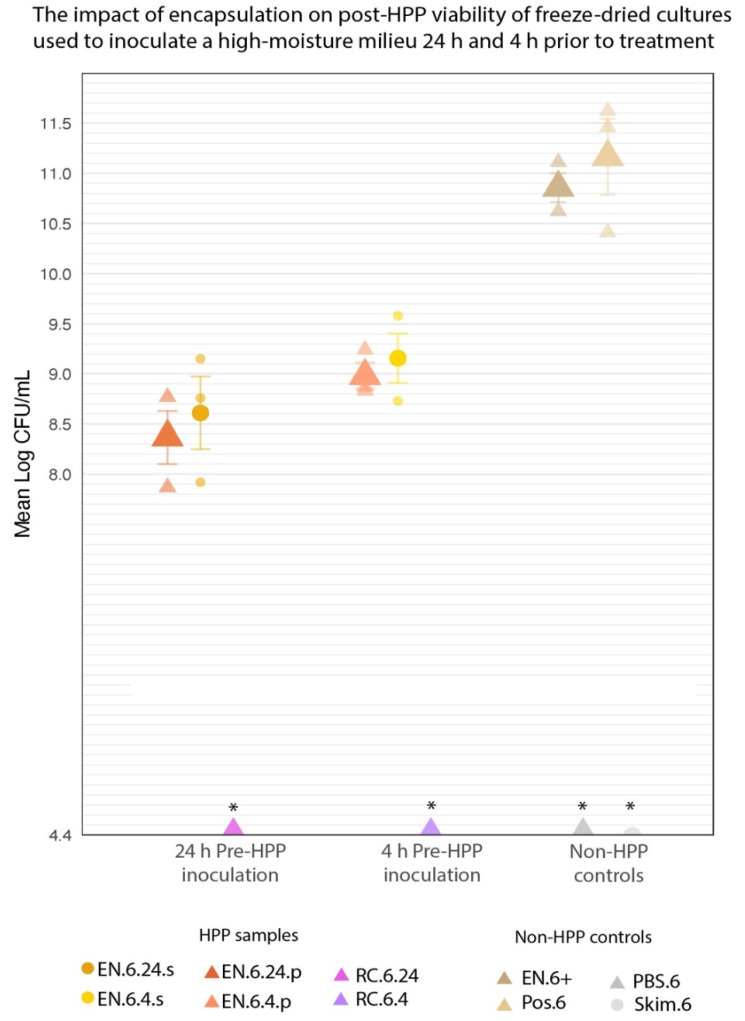
The mean log-transformed counts of encapsulated cultures used for inoculating phosphate-buffered saline or skim milk at either 24 h or 4 h prior to HPP compared to non-HPP treated controls; Pos.6 and EN.6+. Error bars represent SE. * RC-HPP.6 samples and negative controls (PBS.6 and Skim.6) counts were below the level of detection (4.4 log CFU/mL).

**Figure 11 foods-11-02465-f011:**
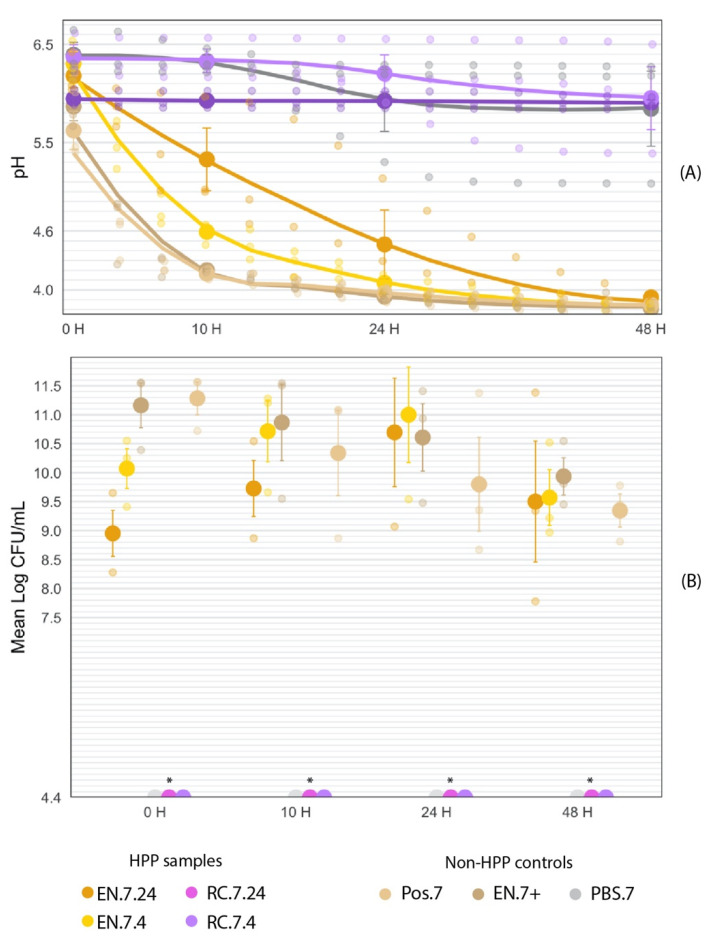
Fermentation (40 °C, 48 H) curve of encapsulated HPP-treated cultures in comparison to non-HPP treated (Pos.7, EN.7+, and MRS.7) and non-encapsulated treated samples (RC.7.24 and RC.7.4). The large circles represent the mean value within each treatment collected at time points 0, 10, 24, and 48 h into the fermentation. Error bars represent SE. (**A**) Fermentation curve of encapsulated HPP-treated cultures in comparison to non-HPP treated (Pos.7, EN.7+, and MRS.7) and non-encapsulated treated samples (RC.7.24 and RC.7.4). (**B**) Mean log-transformed cell counts of encapsulated HPP-treated cultures in comparison to non-HPP treated positive controls (Pos.7, EN.7+) over the course of a 48-h fermentation. * RC-HPP.7 and PBS.7 counts were below the level of detection (4.4 log CFU/mL).

**Table 1 foods-11-02465-t001:** Experiment design for cocoa butter encapsulation with the uncoded values of levels for the three encapsulation parameters: homogenization speed (rpm), homogenization duration (minutes), and cocoa butter temperature at the time of homogenization (°C). Level values are derived from the generated Box-Behnken design (BBD) matrix.

Trial	Speed (rpm) X_1_	Duration (min) X_2_	Temperature (°C) X_3_
1	4000	2	40
2	5000	3	40
3	3000	2	35
4	5000	2	35
5 *	4000	3	35
6	4000	2	30
7	3000	4	35
8	3000	3	40
9	4000	4	30
10	4000	4	40
11	3000	3	30
12	5000	3	30
13 *	4000	3	35
14	5000	4	35

* The Box-Behnken design’s two center points are defined as the mid-level values of the three variables.

**Table 2 foods-11-02465-t002:** The cell counts of the EN-HPP.6treatments displayed significant differences when compared to those of the controls (EN.6+ and Pos.6) but maintained high counts, despite their inoculation in a high-moisture matrix prior to HPP. Listed are the mean log-transformed cell counts and SE obtained for each treatment.

Treatment	Log CFU/mL
EN.6.24.p	8.37 ^a^ ± 0.26
EN.6.4.p	8.98 ^a^ ± 0.13
EN.6.24.s	8.61 ^a^ ± 0.36
EN.6.4.s	9.16 ^a^ ± 0.25
RC.6.24	nd *
RC.6.4	nd *
EN.6+	10.86 ^b^ ± 0.14
Pos.6	11.16 ^b^ ± 0.38
PBS.6	nd *
Skim.6	nd *

^a,b^ Different superscripts indicate significant differences in average cell counts between treatments (*p*-value < 0.05). nd * (not detected): the counts for the RC-HPP.6 samples (RC.6.24 and RC.6. 4) and negative controls (Skim.6 and PBS.6) were not observable at the limit of detection (4.4 log CFU/mL) and were excluded from the statistical analysis.

**Table 3 foods-11-02465-t003:** EN-HPP.7 (EN.7.24 and EN.7.4) samples maintained their acidification activity when compared to non-HPP positive controls, though the drop in pH for EN.7.24 was delayed when compared to EN.7.4 and the positive controls at T = 10 h. Listed are the mean pH values and SE obtained at each time point. Differences in cell counts across the EN-HPP.7 treatments and the controls (EN.7+ and Pos.7) were significant at the beginning of the fermentation but lost their significance around the ten-hour mark. Listed are the mean log-transformed cell counts and SE for each treatment at each time point.

Treatment		T = 0	T = 10	T = 24	T = 48
EN.7.24	pH	6.18 ^a,b^ ± 0.12	5.33 ^a^ ± 0.32	4.36 ^a^ ± 0.35	3.92 ^a^ ± 0.03
	CFU/mL	8.95 ^1^ ± 0.40	9.73 ± 0.48	10.70 ± 0.94	9.50 ± 1.04
EN.7.4	pH	6.30 ^a,b^ ± 0.12	4.59 ^b^ ± 0.07	4.07 ^a^ ± 0.05	3.83 ^a^ ± 0.02
	CFU/mL	10.07 ^1^ ± 0.34	10.71 ± 0.53	11.00 ± 0.83	9.57 ± 0.48
RC.7.24	pH	5.95 ^a,b^ ± 0.048	5.95 ^a,c^ ± 0.051	5.92 ^b^ ± 0.052	5.91 ^b^ ± 0.028
	CFU/mL	nd *	nd *	nd *	nd *
RC.7.4	pH	6.37 ^b^ ± 0.12	6.37 ^c^ ± 0.12	6.20 ^b^ ± 0.19	5.95 ^b^ ± 0.32
	CFU/mL	nd *	nd *	nd *	nd *
MRS.7	pH	6.39 ^b^ ± 0.13	6.32 ^c^ ± 0.07	5.93 ^b^ ± 0.32	5.84 ^b^ ± 0.38
	CFU/mL	nd *	nd *	nd *	nd *
Pos.7	pH	5.87 ^a,b^ ± 0.14	4.20 ^b^ ± 0.01	3.93 ^a^ ± 0.02	3.81 ^a^ ± 0.01
	CFU/mL	11.28 ^2^ ± 0.28	10.34 ± 0.73	9.80 ± 0.81	9.35 ± 0.28
EN.7+	pH	5.62 ^a^ ± 0.20	4.17 ^b^ ± 0.03	3.97 ^a^ ± 0.05	3.85 ^a^ ± 0.04
	CFU/mL	11.16 ^2^ ± 0.39	10.87 ± 0.66	10.61 ± 0.58	9.94 ± 0.32

^a,b,c^ Means within a column with different superscript letters indicate significant differences in pH values between treatments pH during that time point (*p*-value < 0.05). ^1,2^ Means within a column with different superscripts indicate significant differences in counts between treatments during that timepoint (*p*-value < 0.05). nd * (not detected): the counts for the RC-HPP.7 samples (RC.7.24, RC.7.4) and negative control (MRS.7) were not observable at the limit of detection (4.4 log CFU/mL) and were excluded from the statistical analysis.

## Data Availability

The data presented in this study are available on request from the corresponding author.

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
