# Peer review of "Selective Survival of Protective Cultures during High-Pressure Processing by Leveraging Freeze-Drying and Encapsulation"

_foods, 2022, doi:10.3390/foods11162465_

Round 1
Reviewer 1 Report
Dear Editors and authors,
The manuscript (Selective survival of protective cultures during high-pressure 2 processing by leveraging freeze-drying and encapsulation) needs many corrects and modifications.
1- The abstract of the manuscript is weak and needs to add numbers of the results.
2-The introduction of the manuscript needs to add some references (1-(Niamah, A. K., Al-Sahlany, S. T. G., Ibrahim, S. A., Verma, D. K., Thakur, M., Singh, S., ... & Utama, G. L. (2021). Electro-hydrodynamic processing for encapsulation of probiotics: A review on recent trends, technological development, challenges and future prospect. Food Bioscience, 44, 101458.)
2-Nguyen, T. H., Kim, Y., Kim, J. S., Jeong, Y., Park, H. M., Kim, J. W., ... & Kang, C. H. (2020). Evaluating the cryoprotective encapsulation of the lactic acid bacteria in simulated gastrointestinal conditions. Biotechnology and bioprocess engineering, 25(2), 287-292.)
3-This manuscript does not contain a clear goal, a goal must be added that is clear and indicates the purpose of working of the this study.
4-The scientific names of lactic acid bacteria should be written according to the modern nomenclature, such as Lacticaseibacillus paracasei.
5-Propionibacterium freudenreichii subsp. shermanii does not belong to lactic acid bacteria.
In most of the research chapters, the authors refers to lactic acid bacteria, and these bacteria do not belong to it, but rather belong to the propionic acid bacteria.
6-Many of the devices and materials used in the study did not mention their origin like pH meter.
7-Many errors in grammar must be corrected , example see line 159-160.
8-Delete a table 2and 3 These tables are not needed in the manuscript.
9-Page 10, 2.8 Enumeration, How did you calculate the number of bacteria ?write the equation.
10-Enumeration of Propionibacterium freudenreichii subsp. shermanii on MRS agar is big error. The growth of these bacteria on MRS is very weak or non grow.
11-The chapter of working methods is not arranged and organized. The figures 2 and 3 must be modified well. This separation cannot be accepted in the current form.
12-The manuscript contains no conclusions, Why????

Author Response
Response to Reviewer 1 Comments
Point 1
The abstract of the manuscript is weak and needs to add numbers of the results.
Response 1
Thank you for your suggestion. The 200-word count proved challenging when summarizing such a nuanced and complicated study. However, I agree with the reviewer that improvements, like the addition of essential data and improved clarity, could be achieved without sacrificing the brevity of that section. In addition to highlighting the study's results (lines 13-14, 17, 21-23), the abstract was re-written to provide more context to the significance of this work with respect to the current limitations surrounding HPP.
Point 2
The introduction of the manuscript needs to add some references
• Niamah, A. K., Al-Sahlany, S. T. G., Ibrahim, S. A., Verma, D. K., Thakur, M., Singh, S., ... & Utama, G. L. (2021). Electro-hydrodynamic processing for encapsulation of probiotics: A review on recent trends, technological development, challenges and future prospect. Food Bioscience, 44, 101458.
• Nguyen, T. H., Kim, Y., Kim, J. S., Jeong, Y., Park, H. M., Kim, J. W., ... & Kang, C. H. (2020). Evaluating the cryoprotective encapsulation of the lactic acid bacteria in simulated gastrointestinal conditions. Biotechnology and bioprocess engineering, 25(2), 287-292
Response 2
Thank you for suggesting these two bodies of work to strengthen my introduction (line 40). Many of the articles cited in my introduction focus on HPP, so we appreciate your input from the encapsulation and probiotic perspective. Potential readers may find my paper of interest due to the encapsulation aspect rather than the HPP angle, and having additional resources curated with that audience in mind would increase my work's impact.
Point 3
This manuscript does not contain a clear goal, a goal must be added that is clear and indicates the purpose of working of the this study.
Response 3
Thank you for bringing this to our attention. As a corrective action, we have restated the goal at the end of the first paragraph of the introduction (line 35). Additionally, significant revisions to the introduction were implemented to provide a more direct trajectory of how the goal is relevant to the current limitation of HPP. Lastly, we reiterate the goal in the discussion (line 604).
We believe these modifications will help orient the reader and more effectively convey the purpose of the study.
Point 4
The scientific names of lactic acid bacteria should be written according to the modern nomenclature, such as Lacticaseibacillus paracasei.
Response 4
Thank you for bringing this to our attention. When writing the culture section in the methods, we referenced the product spec sheet provided by Danisco and were unaware that the nomenclature was outdated. We corrected the mistake on line 112.
Point 5 & 10 (combined)
Propionibacterium freudenreichii subsp. Shermanii does not belong to lactic acid bacteria. In most of the research chapters, the authors refers to lactic acid bacteria, and these bacteria do not belong to it, but rather belong to the propionic acid bacteria.
Enumeration of Propionibacterium freudenreichii subsp. shermanii on MRS agar is big error. The growth of these bacteria on MRS is very weak or non grow.
Response 5 & 10 (combined)
We agree with the reviewer that Propionibacterium freudenreichii subsp. shermanii is not a LAB. Lactobacillus paracasei is a very common LAB species used in commercial protective culture mixes, so we chose to deliberately select for (and subsequently discuss) the LAB portion of the commercially available product. Our reason for focusing exclusively on LAB is because the novelty of this work is the encapsulation system, not the inter-dynamics of the protective cultures we choose. By using media that selects for LAB, we were able to show that this system is effective at preserving our desired culture. Although we could run a parallel experiment accounting for the metabolic needs of PAB, we believed this would have added an additional layer of complexity to an already complicated study.
Point 6
Many of the devices and materials used in the study did not mention their origin like pH meter.
Response 6
Thank you for your suggestion. Although the pH probe and pH monitor were included in the original manuscript (line 312 in original document, 328 in revised), our in-depth review prompted by this comment revealed that we were missing pertinent information on key elements like the cocoa butter vendor, HPP manufacturer, and R software version.
We have included the missing information accordingly, which can be found on lines 181, 117, and 152, respectively.
If you know about any further mistakes, please let us know in detail.
Point 7
Many errors in grammar must be corrected , example see line 159-160.
Response 7
We thank the reviewer for their comments on the manuscript’s grammar. We have endeavored to correct any mistakes and re-written several sentences throughout the manuscript to improve legibility. Examples include units to lines 159-160 and removed the period from the title of the manuscript.
Additionally, the paper was subjected to an extensive review by a writing tutor from the John S. Knight Institute for Writing in the Disciplines to correct any issues relating to grammar, punctuation, and syntax.
If you know about any other mistakes, please let us know in detail.
Point 8
Delete a table 2and 3 These tables are not needed in the manuscript
Response 8
Thank you for your suggestion. We have removed the unnecessary tables.
Point 9
9-Page 10, 2.8 Enumeration, How did you calculate the number of bacteria ?write the equation.
Response 9
Thank you for your suggestions. We've added a sentence to the body of the manuscript (line 361), detailing the settings we used with the Qcount:
"The following settings were utilized in Qcount’s software: Source: Spread; Minimum size: 0.10mm; Maximum size: 20.00mm; Shutter speed: 1/125; Lighting: bottom; Count light colonies?: True; Reduced region?: True; Grid: circular; Sample volume: 0.1mL; Area multiplier: 5.00%; Low count: 20 colonies; High count: 300 colonies."
Based on the settings selected, for each plate processed, the Qcount program provides two output pieces of data: "count used" and "total count". The "count used" value represents the number of colonies that the Qcount has counted on the respective plate. The "total count" is the final desired value: CFU/mL of the original undiluted sample. This "total count" is calculated by the Qcount program. The Qcount manual does not specify the formula used by the device to calculate the "total count" value; here is the formula that we've created that provides the "total count" value that Qcount computes, based on the appropriate inputs and the "count used" obtained:
(“count used” + “count used” x “area multiplier”/100) x 10”dilution”/”sample volume” = “total count”
We will abstain from including this information in the manuscript, as the values are provided by Qcount, without the need for additional calculations by the researchers. We've included this information here for your reference.
Point 11
The chapter of working methods is not arranged and organized. The figures 2 and 3 must be modified well. This separation cannot be accepted in the current form.
Response 11
We agree with the reviewer the methods section was very confusing, as the experiment was complicated with a lot of very similar samples undergoing minor differences in treatment. Overall, the syntax of the written content was heavily revised in the introduction and methods to improve comprehension under the instruction of the Knight writing center. Additionally, we feel optimistic about the revisions done to our experimental process flow diagram to enhance clarity. Specific improvements include breaking down figure 3 (figure 3-6) by the method's corresponding subsection (2.3-2.7) and including a more visually clear sample glossary. Additionally, the sample names were modified to include the subsection number to help distinguish similar samples between subsections.
As for figure 2, I would benefit from more specific suggestions on how to modify it. We are unsure what you mean by "This separation cannot be accepted in the current form" and would benefit from more specific instructions on improving it.
Point 12
The manuscript contains no conclusions, Why????
Response 12
Thank you for your suggestion. Initially, we were under the impression a conclusion was not required based on the journal's requirements:
Under MDPI's Foods "Instruction for authors":
Conclusions: This section is not mandatory but can be added to the manuscript if the discussion is unusually long or complex.
In the original document, we attempted to combine the conclusion within the discussion section, with the below text serving as the main takeaway (lines 686-688). However, since a conclusion is recommended for complex discussions, we recognized that separating the two sections may improve comprehension. As such, we added a conclusion section where I reiterated the project's main findings and overall take-home message. These changes can be found starting on line 704.

Reviewer 2 Report
Type of the Paper (Article
Foods
Article
Selective survival of protective cultures during high-pressure processing by leveraging freeze-drying and encapsulation
The authors demonstrated the protective effect against high-pressure processing (HPP) provided by the freeze-drying of pressure-sensitive protective cultures and also developed an encapsulation system that maintained the desiccated state of freeze-dried cultures when they were immersed in a high-moisture matrix.
It was also found that a simple cocoa butter encapsulation system for freeze-dried protective cultures can be used in combination with HPP without detrimental effects on cell viability or acidification capacities.
The research design and English throughout the manuscript are very good.
Orientation was clear as required for the audience.
Not many grammatical errors were found. Give one more reading to it.
The manuscript is with clear pictures and also adequate figures and tables.
May change figure 2 for more clarity, the background colour shade is dominating
Figures 6 is with smaller fonts size, authors may think of combining a,b,c, to one or depict it in a more clear-cut way or even changing colours will also be more attractive.
With Regards,
Author Response
We thank the reviewer for taking the time to review our manuscript.
Point 1
Not many grammatical errors were found. Give one more reading to it.
Response 1
We thank the reviewer for their comments on the manuscript’s grammar. We have endeavored to correct any mistakes and re-written several sentences throughout the manuscript to improve legibility. Examples include units to lines 159-160 and removed the period from the title of the manuscript.
Additionally, the paper was subjected to an extensive review by a writing tutor from the John S. Knight Institute for Writing in the Disciplines to correct any issues relating to grammar, punctuation, and syntax.
If you know about any other mistakes, please let us know in detail.
Point 2
May change figure 2 for more clarity, the background colour shade is dominating
Response 2
We agree with the reviewer about the background color and have removed the grey from all the corresponding graphics. We thank the reviewer for this suggestion because we believe it enhances the aesthetic of the visual content.
Point 3
Figures 6 is with smaller fonts size, authors may think of combining a,b,c, to one or depict it in a more clear-cut way or even changing colours will also be more attractive.
Response 3
Although we agree with the reviewer, the response plots generated from the RSM package are not nearly as dynamic as other more common R-based visualization tools (i.e., ggplot2). As such, we cannot modify the text size.
Reviewer 3 Report
The paper titled “Selective survival of protective cultures during high-pressure processing by leveraging freeze-drying and encapsulation”, could represent a valid topic, but in my opinion was written in a confusing and unreadable way, for this reason in this form cannot be accepted.
- I advise the authors to read the manuscript carefully and to review the English form and to follow the guidelines of the Journal
- delete the full stop in the title
- I recommend the authors to modify the abstract section, in this form it is not very understandable.
- I recommend the authors to modify the introduction section, tends to confuse the reader
- Line 55-56, Tsevdou et al. (reference number)
- Line 78. Kurtmann et al. (reference number)
- … Check all the references respecting the guidelines of the Journal.
- I recommend the authors to modify the material and methos section, was developed in a confusion way.
- I advise you to fixed better the tables

Author Response
Point 1
I advise the authors to read the manuscript carefully and to review the English form and to follow the guidelines of the Journal
Response 1
We thank the reviewer for their comments on the manuscript’s grammar. We have endeavored to correct any mistakes and re-written several sentences throughout the manuscript to improve legibility. Examples include units to lines 159-160 and removed the period from the title of the manuscript.
Additionally, the paper was subjected to an extensive review by a writing tutor from the John S. Knight Institute for Writing in the Disciplines to correct any issues relating to grammar, punctuation, and syntax.
If you know about any other mistakes, please let us know in detail.
Point 2
delete the full stop in the title
Response 2
We thank the reviewer for this suggestion and have removed the period from the title.
Point 3
I recommend the authors to modify the abstract section, in this form it is not very understandable.
Response 3
Thank you for your suggestion. The 200-word count proved challenging when summarizing such a nuanced and complicated study. However, I agree with the reviewer that improvements, like the addition of essential data and improved clarity, could be achieved without sacrificing the brevity of that section. In addition to highlighting the study's results (lines 13-14, 17, 21-23), the abstract was re-written to provide more context to the significance of this work with respect to the current limitations surrounding HPP.
Point 4
I recommend the authors to modify the introduction section, tends to confuse the reader
I recommend the authors to modify the material and methos section, was developed in a confusion way.
Response 4
Thank you for bringing this to our attention. The syntax of the written content was heavily revised in the introduction and methods sections to improve comprehension under the instruction of the Knight writing center.
In the introduction, we have restated the goal at the end of the first paragraph (line 35). Additionally, significant revisions to the introduction were implemented to provide a more direct trajectory of how the goal is relevant to the current limitation of HPP. We believe these modifications will help orient the reader and more effectively convey the purpose of the study.
Additionally, we feel optimistic about the revisions done to our experimental process flow diagram to enhance clarity. Specific improvements include breaking down figure 3 (figure 3-6) by the method's corresponding subsection (2.3-2.7) and including a more visually clear sample glossary. Another major revision to the methods was the modifications to the sample names, which now include the subsection number to help distinguish similar samples between subsections.
Point 5
Line 55-56, Tsevdou et al. (reference number)
Line 78. Kurtmann et al. (reference number)
Response 5
We thank you for your suggestion and have correctly implemented the author’s citations on lines 48 and 76, respectively.
Point 6
… Check all the references respecting the guidelines of the Journal.
Response 6
We thank you for your input and have thoroughly reviewed the journal’s guidelines. We believe the revised manuscript should be compliant with it.
Point 7
I advise you to fixed better the tables
Response 7
We thank you for your suggestion. We have modified the tables to include SE on the same cell as the data value being collected (i.e., CFU/mL or pH).
If you have any other suggestions regarding the data tables, please let us know in detail.
Round 2
Reviewer 1 Report
Dear Editor,
The authors made all the necessary changes to improve the manuscript, and now I recommend it for publication in its current form.
Reviewer 3 Report
The authors have modified the manuscript according to my personal observations. In my opinion the paper titled: “Selective survival of protective cultures during high-pressure processing by leveraging freeze-drying and encapsulation”, can be consider for the publication in this Journal.
I advise authors to make only a few changes such as:
- Fig.3.4,5,6, 9 reduce the caption character and add a space between the caption and text.